# Upper-Limb Functional Recovery in Chronic Stroke Patients after COVID-19-Interrupted Rehabilitation: An Observational Study

**DOI:** 10.3390/jcm13082212

**Published:** 2024-04-11

**Authors:** Daigo Sakamoto, Toyohiro Hamaguchi, Yasuhide Nakayama, Takuya Hada, Masahiro Abo

**Affiliations:** 1Department of Rehabilitation Medicine, The Jikei University School of Medicine Hospital, Tokyo 105-8471, Japan; daigo.0612@jikei.ac.jp; 2Department of Rehabilitation, Graduate School of Health Science, Saitama Prefectural University, Saitama 343-8540, Japan; 3Department of Rehabilitation Medicine, The Jikei University School of Medicine, Tokyo 105-8461, Japan; pt_nakayama@jikei.ac.jp (Y.N.); t-hada@jikei.ac.jp (T.H.)

**Keywords:** upper extremity, motor paralysis, occupational therapy, outpatient, stroke, COVID-19, activities of daily living

## Abstract

**Background/Objectives**: Upper-limb function of chronic stroke patients declined when outpatient rehabilitation was interrupted and outings restricted, owing to the novel coronavirus infection (COVID-19) pandemic. We investigated whether these patients recovered upper-limb function post-resumption of outpatient rehabilitation. **Methods**: In this observational study, 43 chronic stroke hemiparesis patients with impaired upper extremity function were scored for limb function via the Fugl-Meyer assessment of the upper extremity (FMA-UE) and the Action Research Arm Test (ARAT) after a structured interview, evaluation, and intervention. Scores at 6 and 3 months pre- and 3 months post-rehabilitation interruption were examined retrospectively; scores immediately and at 3 and 6 months post-resumption of care were examined prospectively. The amount of change for each time period and an analysis of covariance were performed with time as a factor, changes in the FMA-UE and the ARAT scores as dependent variables, and statistical significance at 5%. **Results**: The time of evaluation significantly impacted the total score, as well as part C and part D of FMA-UE and total, pinch, and gross movement of the ARAT. Post-hoc tests showed that the magnitude of change in limb-function scores from immediately to 3 months post-resumption was significantly higher than the change from 3 months pre- to immediately post-interruption for the total score and part D of the FMA-UE, as well as grip and gross movement of the ARAT (*p* < 0.05). **Conclusions**: Upper-limb functional decline in chronic stroke patients, caused by the COVID-19 pandemic-related therapy interruption and outing restrictions, was resolved approximately 3 months post-resumption of rehabilitation therapy. Our data can serve as reference standards for planning and evaluating treatment for chronic stroke patients with inactivity-related impaired upper-limb function.

## 1. Introduction

The measures implemented to control a pandemic of novel coronavirus infection (COVID-19) caused by severe acute respiratory syndrome coronavirus 2 (SARS-CoV-2) appear to have had a negative impact on the health of some people. COVID-19 spread rapidly after its outbreak in December 2019, and the World Health Organization declared it a pandemic [1]. Lockdowns were implemented around the world to prevent the spread of SARS-CoV-2, and a state of emergency was declared in Japan [2,3,4]. People were forced to change their lifestyles, as non-essential outings were restricted. People spent more time at home, and this led to longer sedentary hours and less physical activity during the day [5]. The decrease in physical activity in older adults and patients receiving physiotherapy caused physical and mental harm [6,7]. 

Post-stroke motor paralysis limits patient activities of daily living (ADL) and reduces their quality of life [8,9]. Rehabilitation treatment is important to maintain and improve ADL and the quality of life of stroke patients [10]. However, during the COVID-19 pandemic, it became challenging to provide effective rehabilitation treatment for stroke patients [11]. Although telerehabilitation is becoming more widespread, it was not developed in time during the pandemic, resulting in limited opportunities for chronic stroke patients to receive treatment [12]. It has been reported that COVID-19-related interruption in care for patients with hemiparesis after chronic stroke can lead to the worsening of upper-limb motor function and subjective physical symptoms [13]. If this temporary decline in physical functionality can be reversed with subsequent support, the importance of continued rehabilitation will be recognized.

The purpose of this study was to estimate the amount of recovery of upper-limb function in chronic stroke patients after COVID-19-related interruption of outpatient care. In this study, chronic stroke patients we had previously studied [13] were re-investigated after a practice plan was developed and rehabilitation was resumed within the scope of usual practice. The hypothesis of this study was that the resumption of outpatient rehabilitation improves upper-limb motor function in patients with hemiparesis due to chronic stroke, whose upper-limb motor function had declined as a result of interrupted outpatient rehabilitation and refraining from going outside. Since functional prognosis may differ between stroke and cerebral hemorrhage [14,15], another hypothesis was that the course of recovery of upper-limb function after the resumption of rehabilitation would vary according to stroke type. The results of this study can be used as a reference for future exercise planning and for devising ways to prevent the decline in upper-limb function in chronic stroke patients.

## 2. Materials and Methods

### 2.1. Study Design

This was an observational study on pre- and post-survey data without a control group.

### 2.2. Participants

Our study included patients who had received outpatient occupational therapy for at least 6 months at the Department of Rehabilitation Medicine of the Jikei University Hospital between 1 June 2019 and 31 May 2020, were post-stroke hemiplegic patients for whom at least 6 months had passed since onset, were at least 20 years of age, and were patients whose outpatient rehabilitation was temporarily interrupted due to the spread of SARS-CoV-2 infection. Patients were excluded if they had a diagnosis of higher brain dysfunction, cognitive impairment, or a psychiatric disorder that would affect functional assessment measures and understanding of instructions during rehabilitation. After confirming that the patients met the eligibility criteria, we provided written and oral explanations of the study and requested their participation. Those who agreed to participate were considered eligible for the study.

The minimum sample size was calculated to be 40 cases in total using G*Power 3.1 software, with a goodness of fit test (F-test) and analysis of variance (for fixed effects, omnibus, one-way ANOVA), effect size f = 0.73, α = 0.05, power = 0.95, number of groups = 4, numerator df = 3, and partial η^2^ = 0.35. A one-way ANOVA is a statistical method used to compare the means of three or more samples using the F-distribution, where α is the significance level and the probability value that serves as a criterion to reject the null hypothesis. 

### 2.3. Survey Periods and Instruments

This study was conducted approximately 6 months before outpatient rehabilitation was interrupted (6 m before), approximately 3 months before outpatient rehabilitation was interrupted (3 m before), after the interruption period (after IP), approximately 3 months after outpatient rehabilitation was resumed (3 m after), and approximately 6 months after outpatient rehabilitation was resumed (6 m after). The date on which outpatient occupational therapy was suspended due to the SARS-CoV-2 infection outbreak was 1 April 2020. At 6 m before and 3 m before, the Fugl-Meyer’s assessment of the upper extremity (FMA-UE) and Action Research Arm Test (ARAT) scores were examined retrospectively from the medical records. At after IP, 3 m after, and 6 m after, FMA-UE and ARAT scores were prospectively assessed (Figure 1).

### 2.4. Occupational Therapy for Outpatients

Occupational therapists conducted structured interviews, assessments, and interventions within the usual scope of practice with patients who had resumed outpatient rehabilitation (Table 1). The questionnaire used during the interviews with patients was developed based on the International Classification of Functioning, Disability, and Health (ICF) category [16] and comprised questions regarding function, activity, and participation (Figure 2).

The occupational therapist shared a practice plan with the patient that had been developed with consideration for the subjective symptoms that occurred by the patient and provided practice and instruction. The occupational therapists identified items with decreased ratings in the upper extremity functionality of the patient that were a high priority for treatment, described exercises to improve limb function, and guided the patient in self-training. The main types of exercises included a joint range of motion exercises to improve joint contractures, stretching exercises for spastic muscles to decrease muscle tension, and upper-limb function exercises to improve motor paralysis (Appendix A). Pamphlets containing the instructions were distributed to the patients. The instructions were modified or added to as the subjective symptoms and upper extremity functionality of the patients changed. In the second and subsequent sessions, the occupational therapist provided positive feedback to the patient on the performance of self-training and changes in limb function and movement to maintain and improve patient motivation.

### 2.5. Main Outcome

The primary outcome was the change in FMA-UE scores [17]. The FMA-UE scores were classified according to the report of Woodbury et al. to determine the severity of motor paralysis [18].

### 2.6. Secondary Outcome

The secondary outcome was the change in ARAT scores. The ARAT is an upper extremity functional assessment developed based on the upper extremity function test [19].

### 2.7. Participant Characteristics

The characteristics of the subjects were investigated in terms of age, gender, height, weight, body mass index (BMI), dominant hand, and Barthel Index [20]. As part of patient information, the type of stroke, the side of paralysis, the duration since the onset of stroke, the comorbidities (including myocardial infarction, congestive heart failure, chronic lung disease to the extent that dyspnea occurs on exertion, diabetes mellitus, and renal dysfunction to the extent that dialysis or kidney transplantation is required), whether or not botulinum treatment was given, the duration of discontinuation of outpatient rehabilitation, the duration at each evaluation point, the number of occupational therapy sessions per month, and the duration per session were investigated.

### 2.8. Investigators

Investigations of upper-limb motor function, ADL, and treatment of patients were conducted by seven occupational therapists working at Jikei University Hospital and engaged in rehabilitation in the area of cerebrovascular disorders for more than 5 years. The correlation test was performed between these therapists, which confirms a similar application of the tests.

### 2.9. Statistical Analysis

The total scores of FMA-UE and ARAT and the scores of the sub-items were calculated as the amount of change (delta) using the following equation to estimate their recovery: Date period 1 = 3 m before—6 m before score; (1)
Date period 2 = after IP—3 m before score;(2)
Date period 3 = 3 m after—after IP score;(3)
Date period 4 = 6 m after—3 m after score. (4)

To test the hypothesis that the resumption of outpatient rehabilitation improves upper-limb motor function in patients with hemiparesis due to chronic stroke, whose upper-limb motor function had declined as a result of interrupted outpatient rehabilitation and refraining from going outside, we used time as a factor and changes in FMA-UE and ARAT as dependent variables. An analysis of covariance (ANCOVA) was conducted. ANCOVA is a statistical method used to compare differences in means after adjusting for the effects of covariates that are assumed to affect the dependent variable in addition to the independent variables. The covariates were age, gender, BMI, time since onset, and the 6 m before score [21,22]. A two-way ANCOVA was performed as a subanalysis to examine the amount of recovery of upper-limb function by stroke type (cerebral infarction, cerebral hemorrhage). The dependent variables and covariates were set as in the main analysis, with time of evaluation and stroke type used as factors. JASP 0.16 software (University of Amsterdam Department of Psychology & Psychological Methods Unit, Amsterdam, The Netherlands) was used for the statistical analysis. A *p*-value of <0.05 was considered statistically significant.

### 2.10. Ethical Considerations

All patients who participated in the study gave written consent to participate in the study. The study was conducted in accordance with the guidelines of the Declaration of Helsinki [23] and was approved by the Ethics Committee of the Jikei University School of Medicine (Approval Number 24-295-7061).

## 3. Results

Between 1 June 2019 and 31 May 2020, 81 patients with chronic stroke hemiparesis were enrolled in our study, undergoing at least 3 months of outpatient rehabilitation. Of 81 patients, 73 patients met the eligibility criteria, and 49 patients were included in the study. It has been reported by us previously that the upper-limb functions of the patients deteriorated, and subjective physical symptoms occurred after approximately 3 months of interruption of outpatient occupational therapy [13]. In the present study, we followed up on these 49 patients. Of the 49 patients, 6 were excluded; the total number of patients included in the final analysis of the present study was 43 (Figure 3). The characteristics of the patients included in the present analysis are summarized in Table 2. The paralyzed and dominant side of all patients was the right side. One patient had a history of myocardial infarction, and two had a diagnosis of diabetes mellitus. All patients were receiving treatment with botulinum neurotoxins, which were last administered on an average of 1 month before occupational therapy intervention was discontinued. The patients’ interruption period (months) (median (25th, 75th percentile)) was 3 (3). The period from 6 m before to the start date of formal interruption (months) was 7 (5, 9), the period from 3 m before to the start date of formal interruption (months) was 3 (2, 4), the period from after IP to 3 m after (months) was 3 (2, 3), and the period from after IP to 6 m after (months) was 6 (5, 6). The participants’ frequency of occupational therapy per month (median (25th, 75th percentile)) was 1 (1,2), and occupational therapy intervention time per session (minutes) was 60 (40, 60). The FMA-UE and ARAT scores during the study period are shown in Table 3 and Appendix B. ANCOVA with the time of evaluation as a factor and the change in FMA-UE and ARAT scores as the dependent variables showed that the total score (F = 6.925, *p* < 0.001, η^2^ = 0.109), part C (F = 8.458, *p* < 0.001, η^2^ = 0.131), and part D (F = 11.903, *p* < 0.001, η^2^ = 0.178) of FMA-UE, the total score (F = 5.378, *p* = 0.001, η^2^ = 0.116), grip (F = 4.357, *p* = 0.006, η^2^ = 0.074), pinch (F = 11.685, *p* < 0.001, η^2^ = 0.176), and gross movement (F = 7.511, *p* < 0.001, η^2^ = 0.116) of the ARAT showed a significant main effect of time of evaluation. There was no main effect of the evaluation period for part A (F = 2.133, *p* = 0.098, η^2^ = 0.037) and part B (F = 1.867, *p* = 0.137, η^2^ = 0.032) of the FMA-UE test, as well as for grasp (F = 1.192, *p* = 0.315, η^2^ = 0.021) of the ARAT. A two-way ANCOVA as a subanalysis showed a significant interaction of ARAT pinch between the time of evaluation and stroke type (F = 3.2233, *p* = 0.024, η^2^ = 0.046). There was no interaction between the evaluation period and stroke types for the total score (F = 0.296, *p* = 0.828, η^2^ = 0.005), part A (F = 0.636, *p* = 0.593, η^2^ = 0.011), part B (F = 0.816, *p* = 0.487, η^2^ = 0.014), part C (F = 1.987, *p* = 0.118, η^2^ = 0.031), and part D (F = 1.950, *p* = 0.124, η^2^ = 0.028) of the FMA-UE test, as well as for the total score (F = 0.848, *p* = 0.470, η^2^ = 0.014), grasp (F = 0.118, *p* = 0.949, η^2^ = 0.002), grip (F = 1.681, *p* = 0.173, η^2^ = 0.027), and gross movement (F = 2.396, *p* = 0.071, η^2^ = 0.033) of the ARAT. 

Values are mean ± Std. deviation (n = 43). IP, interruption; FMA-UE, Fugl-Meyer assessment of the upper extremity; ARAT, Action Research Arm Test; 6 m before, approximately 6 months before outpatient rehabilitation was interrupted; 3 m before, approximately 3 months before outpatient rehabilitation was interrupted; after IP, after interruption period; 3 m after, approximately 3 months after outpatient rehabilitation was resumed; 6 m after, approximately 6 months after outpatient rehabilitation was resumed; (1) the amount of change in scores from 6 months prior to interruption to 3 months prior to interruption; (2) the amount of change in scores from 3 months before interruption to immediately after resumption; (3) the amount of change in scores from immediately after resumption to 3 months after resumption; (4) the amount of change in scores from 3 months to 6 months after resumption.

ANCOVA showed a significant main effect, and Tukey’s post-hoc test was performed for the items for which a significant main effect was found. In a total of FMA-UE, the change (3) from immediately after resumption to 3 months after resumption was significantly higher than the change (2) from 3 months before interruption to immediately after resumption (mean difference = 2.79, 95% CI lower = 0.92, upper = 4.66, SE = 0.72, t = 3.88, Cohen’s d = 0.66, *p* < 0.001, Figure 4a). Similarly, the change (4) from 3 to 6 months after resumption was significantly higher (mean difference = 2.28, 95% CI lower = 0.41, upper = 4.15, SE = 0.72, t = 3.17, Cohen’s d = 0.64, *p* = 0.010, Figure 4a). In part D (coordination/speed) of FMA-UE, the change (3) from immediately after resumption to 3 months after resumption was significantly higher than the change (2) from 3 months before to immediately after suspension (mean difference = 0.51, 95% CI lower = 0. 01, upper = 1.01, SE = 0.19, t = 2.65, Cohen’s d = 0.42, *p* = 0.043, Figure 4b). In the grip of ARAT, the change (3) from immediately after resumption to 3 months after resumption was significantly higher than the change (2) from 3 months before to immediately after suspension (mean difference = 0.61, 95% CI lower = 0.02, upper = 1.20, SE = 0.23, t = 2.66, Cohen’s d = 0.45, *p* = 0.042, Figure 4c). In the gross movement of ARAT, the change (3) from immediately after resumption to 3 months after resumption was significantly higher than the change (1) from 6 months before interruption to 3 months before interruption (mean difference = 0.46, 95% CI lower = 0.18, upper = 0.75, SE = 0.11, t = 4.18, Cohen’s d = 0.74, *p* < 0.001, Figure 4d). Similarly, it was significantly higher than the change (2) from 3 months before interruption to immediately after resumption (mean difference = 0.40, 95% CI lower = 0.11, upper = 0.69, SE = 0.11, t = 3.55, Cohen’s d = 0.67, *p* = 0.003, Figure 4d). The change (4) from 3 to 6 months after reopening was significantly lower than the change (3) from immediately after reopening to 3 months after reopening (mean difference = −0.33, 95% CI lower = −0.62, upper = −0.04, SE = 0.11, t = −2.92, Cohen’s d = −0.66, *p* = 0.021, Figure 4d). At the time of evaluation, there were no significant differences in the part C scores of FMA-UE and the total and pinch scores of ARAT. A Tukey’s post-hoc test was performed for the pinch of ARAT, which exhibited a significant interaction in the two-way ANCOVA. The result showed no significant differences. 

## 4. Discussion

This study showed that chronic stroke patients with hemiparesis, whose upper-limb function had declined due to the interruption of outpatient rehabilitation and refraining from going out during the SARS-CoV-2 infection outbreak, regained their upper-limb function scores 3 months after rehabilitation was resumed. We have previously reported that upper-limb function deteriorates in chronic stroke patients after 3 months of outpatient rehabilitation and refraining from going out [13]. Our present finding, that these patients regained upper-limb function after rehabilitation, suggests that the effects of temporary physical inactivity can be reversed in approximately 3 months if appropriate rehabilitation is resumed. Although the amount of physical activity varies from person to person, refraining from going out to prevent SARS-CoV-2 infection is thought to have reduced the amount of activity in chronic stroke patients, causing disuse of limbs. In healthy older subjects, 10 days of complete rest results in a 30% decrease in muscle protein synthesis and a 16% decrease in muscle strength [24]. Since it has been reported that stroke patients have less muscle mass than healthy subjects, and paralyzed side limbs of stroke patients have less muscle mass than non-paralyzed side limbs [25,26,27], it is inferred that motor paralysis and decreased activity have a combined impact on the limbs of the paralyzed side in stroke patients, leading to progressive disuse. Brain and nerve changes are due to disuse-dependent plasticity [28]. Disuse syndrome in stroke patients includes short-term effects on the brain and nervous system, as well as slowly progressive effects on the musculoskeletal system over time. The patients in this study had experienced a chronic stroke, and inactivity during the period of interrupted rehabilitation was assumed to have caused a transient decline in upper-limb motor function, affecting both the brain and the motor system.

The results of the present study suggest that the decline in upper-limb motor function due to the interruption of outpatient rehabilitation is a reversible phenomenon. The recovery of function after 3 months of inactivity is attributed to the fact that the interview, assessment, and intervention procedures performed on the patients, within the usual scope of care, promoted use-dependent plasticity in patients. Use-dependent plasticity is a phenomenon in which repeated activation of certain neurons facilitates the same pattern of activity [29,30]. The amount of sufficient practice needed to restore motor paralysis is estimated to be at least 20 h over a 2-week period of intensive practice, and at least 60 h of practice are needed to further enhance the use of the paralyzed upper extremity [31,32,33]. The participants in this study received approximately three sessions of outpatient rehabilitation during the first 3 months after resumption of medical care, with each session lasting approximately 40 min. Although the amount of physical activity in and out of the home also influenced the patients’ recovery of upper-limb motor function, it is likely that the instruction provided by the therapist contributed in part to the recovery of upper-limb motor function. The results of this study confirm that therapist-led instruction, even with limited intervention opportunities, was instrumental in increasing patients’ upper extremity movement on the affected side. Recovery of motor paralysis is facilitated when patients are aware of their own goals and carry out physical exercises independently, and when they are given feedback by the therapist [34,35]. Treatment goal setting, amount of practice, and patient motivation are important for recovery of motor paralysis after stroke [36,37,38]. The assessments made after the resumption of outpatient rehabilitation were shared individually with the therapists and patients, and the patients were made aware of the changes that had occurred in their physical function and their future treatment goals. This was assumed to motivate the patients to engage in the recommended physical exercises. Activity monitors can measure a patient’s upper extremity use [39,40], and low-frequency therapy devices and vibration stimulation devices can reduce spasticity in patients [41,42]. These can be effective tools to improve patient motivation and support independent practice. When the therapist encourages the patient to exercise, information about the loss from not exercising and the gain from carrying it out impacts the patient according to the personality, psychological state, and social circumstances of the patient [43,44]. The results of this study can be used to assist patients in increasing the amount of movement of their paralyzed upper extremities. If the patient does not practice the recommended activities voluntarily or does not increase the use of their hands in daily life, it can serve as information about the loss of motor function and the risk of further deterioration of ADLs. If the patient hesitates to carry out physical activities when rehabilitation can be resumed, this period of inactivity can provide information about gains that can facilitate recovery. By intensifying practice aimed at improving decreased function and movement, as well as increasing the use of the paralyzed upper extremity in ADL, recovery can be facilitated. In the event of an outbreak of an unknown infectious disease such as COVID-19, the data obtained in this study will be useful to therapists who assist patients with exercise and activity, as well as to patients who wish to maintain upper extremity function. In addition, this study will provide useful data for the development of practice programs for the prevention of exacerbation and recovery of upper-limb function and will contribute to the development of effective telerehabilitation treatment methods.

In this study, recovery of upper-limb function was suggested by improvements in part D scores of the FMA-UE, as well as grip and gross movement scores of the ARAT. Patients with more severe motor paralysis showed greater recovery in the ARAT grip and gross movement scores, whereas those with less severe motor paralysis were more likely to improve the quality of life activities using the paralyzed upper extremity when assessed during a 2-week treatment combining transcranial magnetic stimulation and occupational therapy [45]. Reports examining the degree of difficulty in the sub-items of the FMA-UE have shown that the items in part D are more difficult [46,47]. In patients with mild disease, occupational therapists encouraged the use of the paralyzed upper limb for ADL, which presumably restored the smoothness and speed of the part D movements. On the other hand, in patients with severe disease, the range of reach of the paralyzed upper limb was increased by exercises to improve the joint range of motion and muscle flexibility of the paralyzed upper limb. As the opportunity to use the paralyzed upper limb as an aid in sitting movements and to manipulate objects on a desk increased, the ARAT scores for grasp and gross movement may have also improved. The joint range of motion exercises and stretching exercises increased the amount of movement of the paralyzed upper limb and restored the patient’s functional disability due to learned non-use [48], which may have been induced by decreased activity. The ADL exercises provided helped the patient break free from learned bad use of the paralyzed limb [49], reminded him of the correct use of the paralyzed limb, and improved activity limitation. The FMA-UE and ARAT assess different aspects of upper extremity function [50]. The FMA-UE has features that reflect psychosomatic function and body structure and assesses range, accuracy, and segmentation of movement without compensatory strategies. Improvements in FMA-UE scores suggest true recovery of upper extremity motor function rather than an intensification of the patients’ compensatory strategies. The ARAT has characteristics that reflect activity and assesses both motor recovery and compensatory strategies. Improvements in ARAT scores suggest that patients can perform movements that may involve compensatory mechanisms. It is assumed that these upper extremity functional assessments detected a true recovery of motor function and changes in compensatory strategies to improve ADLs brought about by the self-training provided to the patients. In a subanalysis, there was no difference in the amount of recovery of upper-limb function by stroke type. Differences in functional outcomes between patients with cerebral hemorrhage and cerebral infarction have been reported [51]. On the other hand, when intensive rehabilitation was provided to patients in the chronic phase after more than 6 months from the onset of stroke, with induced neuroplasticity of the brain, there was no difference according to stroke type, and both groups reported improved upper-limb function [52]. The present study, similar to the aforementioned one, included patients in the chronic phase, and it was presumed that upper-limb function was restored regardless of stroke type.

This study has some limitations. First, this is an observational study with no control group; thus, there are limitations in asserting that the improvement in upper-limb motor function was solely caused by restarting rehabilitation treatment. A prospective intervention study with a control group will test the additive effect of rehabilitation on the recovery of upper-limb function. Second, the amount of physical activity of patients during the period when rehabilitation was interrupted and the amount and duration of self-training after rehabilitation resumed were not investigated. The recovery of patients’ upper-limb function may depend on the amount of activity in and out of the home, other than self-training supervised by the therapist. The amount of activity is a factor that influences the maintenance of physical function, and we hypothesized that refraining from going out to prevent SARS-CoV-2 infection was a behavior that decreased the amount of activity in patients with chronic stroke. The extent to which the activity level of chronic stroke patients changed during and after the interruption of outpatient rehabilitation should be investigated using activity meters or similar devices when designing new studies in the future. Third, this study did not examine the nutritional status and sleep duration of the patients. These are factors that affect the neuroplasticity of the brain, and this provides the neural basis of use-dependent plasticity [53]. Fourth, because the participants in this study were generally independent in their daily lives, and no patients had severe comorbidities, it is not possible to use the results of this study to estimate the extent of upper-limb function recovery in scenarios involving severely ill patients requiring assistance in daily activities or patients with ADL limitations due to comorbidities that interfere with self-training. Patients with more severe sequelae may have been more susceptible to the COVID-19 pandemic [54]. An analysis by another study with a larger sample size is needed to clarify the amount of upper extremity function recovered by the severity of the illness. Fifth, all patients in this study were treated with botulinum neurotoxins. Spasticity in the patients’ upper extremities limits the recovery of upper extremity motor function and ADLs [55]. However, the effects of these administrations and changes in spasticity on the recovery of upper-limb function have not been clarified. Sixth, questionnaires were used to assess patients’ subjective symptoms, but the reproducibility of these results has not been confirmed. Seventh, since outpatients were included in this study more than 6 months after the onset of illness, it was not possible to obtain information during the acute phase of treatment. The impact of the patients’ acute treatment, length of hospitalization, and complications during hospitalization on the results of this study is not known. Finally, because this study was conducted at a university hospital in Tokyo, Japan, there are limitations to the generalizability of the study in estimating the amount of recovery of upper-limb function for patients residing in other geographical regions, where access to public transportation and the extent of hospital facilities may be very different from this patient cohort.

## 5. Conclusions

The results of this study suggest that the loss of upper-limb function in chronic stroke patients caused by the restriction of outings and interruption of outpatient care due to the spread of SARS-CoV-2 infection was reversible and that upper-limb function was restored 3 months after the resumption of rehabilitation. The temporary functional losses incurred by the patient and the subsequent recovery values can serve as reference standards for the rehabilitation of a patient who has had a period of inactivity of approximately 3 months.

## Figures and Tables

**Figure 1 jcm-13-02212-f001:**
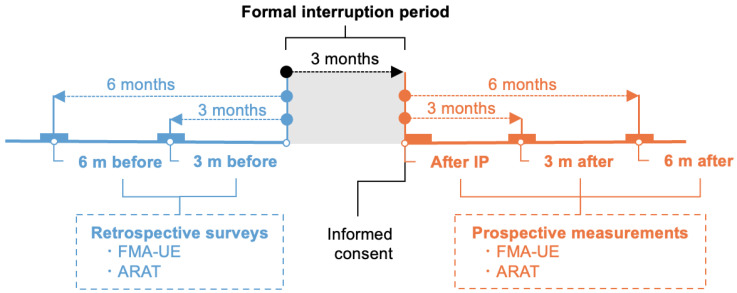
Schematic showing the timing of limb function surveys in relation to the therapy interruption period: 6 m before, approximately 6 months before outpatient rehabilitation was interrupted; 3 m before, approximately 3 months before outpatient rehabilitation was interrupted; after IP, after the interruption period; 3 m after, approximately 3 months after outpatient rehabilitation was resumed; 6 m after, approximately 6 months after outpatient rehabilitation was resumed; FMA-UE, Fugl-Meyer assessment of the upper extremity; ARAT, Action Research Arm Test.

**Figure 2 jcm-13-02212-f002:**
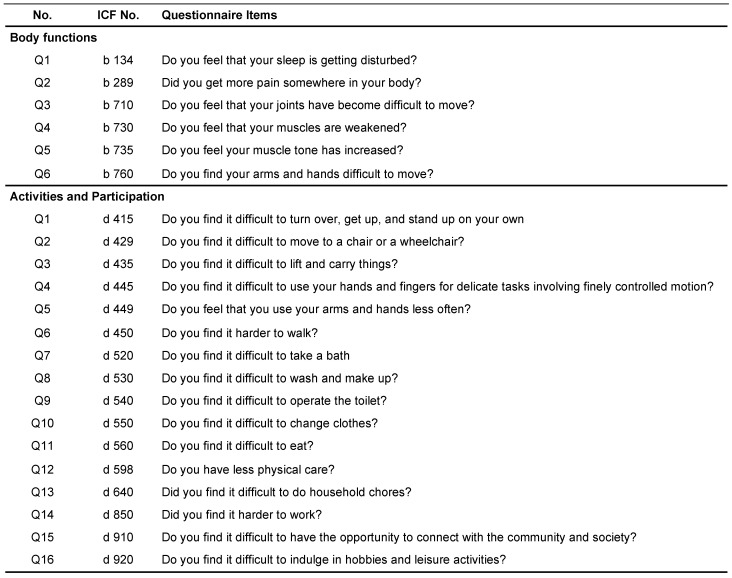
Questionnaire used during interviews with patients. This figure is the same as the one shared in our previous publicaton [13]. ICF, International Classification of Functioning, Disability, and Health.

**Figure 3 jcm-13-02212-f003:**
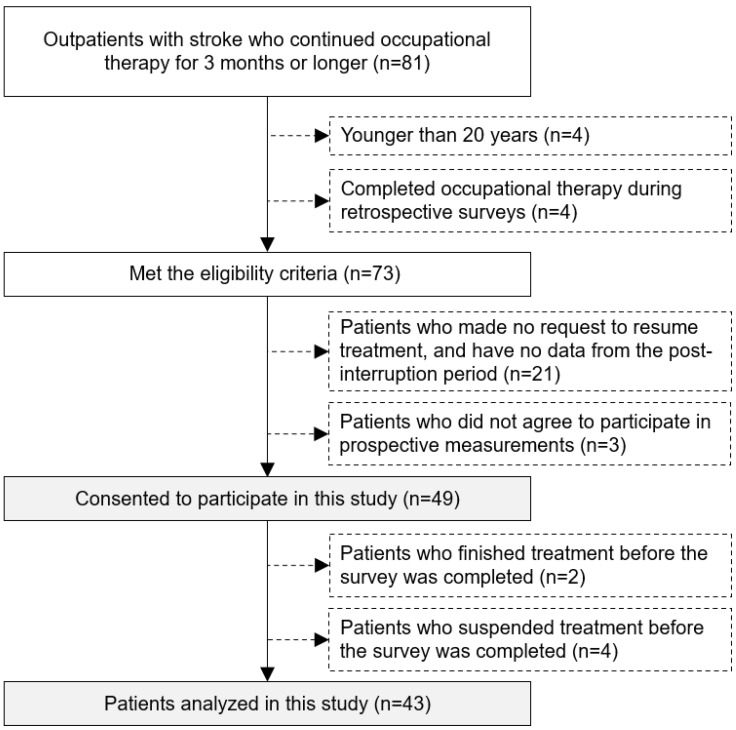
Flow chart showing patient inclusion criteria.

**Figure 4 jcm-13-02212-f004:**
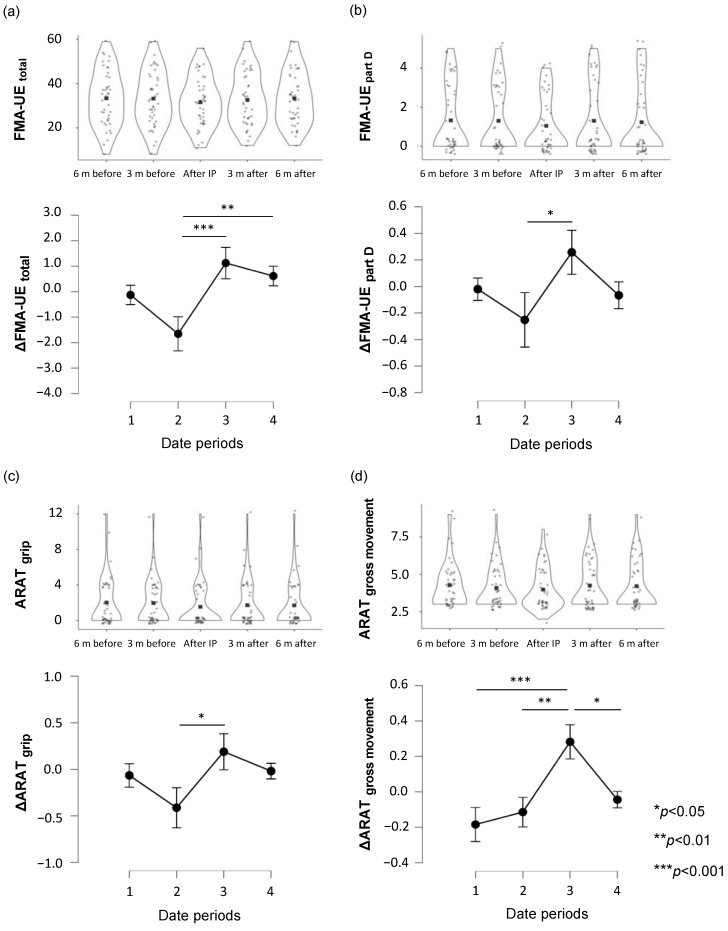
Changes in upper-limb motor function over time. Scores at five-time points are shown in the upper panel, and changes in scores are illustrated in the lower panel for (**a**) FMA-UE total, (**b**) FMA-UE part D, (**c**) ARAT grip, and (**d**) ARAT gross movement. Violin plots are displayed using a kernel density estimator; the shape at each value of the x-axis represents the density of data points corresponding to that score. The black squares in the graph indicate the mean. The gray circles indicate the values for each patient. Statistical significance was set at *p* < 0.05 for Tukey’s multiple comparisons (n = 43). FMA-UE, Fugl-Meyer assessment of the upper extremity; ARAT, Action Research Arm Test; 6 m before, approximately 6 months before outpatient rehabilitation was interrupted; 3 m before, approximately 3 months before outpatient rehabilitation was interrupted; after IP, after interruption period; 3 m after, approximately 3 months after outpatient rehabilitation was resumed; 6 m after, approximately 6 months after outpatient rehabilitation was resumed; date period 1: from 6 months before interruption to 3 months before interruption; date period 2: from 3 months before interruption to immediately after resumption; date period 3: from immediately after resumption to 3 months after resumption; date period 4: from 3 months after resumption to 6 months after resumption.

**Table 1 jcm-13-02212-t001:** Structured interviews, assessments, and interventions performed within the usual scope of practice for patients who resumed outpatient rehabilitation.

No.	Contents
1. Subjective symptoms of patients	
1.1.	Interview the patient about subjective symptoms related to functional disability, activity limitations, and participation restrictions (Figure 2). Establish clear treatment goals based on the problems faced by the patient.
1.2.	The therapist shares the treatment plan with the patient and provides practice and guidance.
2. Upper-limb motor function	
2.1.	The upper-limb functional assessment (FMA, ARAT) is performed. The joint range of motion and muscle tone are assessed. The acquired evaluation values are compared with those in the past, and changes in scores are checked.
2.2.	Practice and instruction on upper extremity function with decreased ratings are provided.
3. Use and practice of the upper limb on the paralyzed side	
3.1.	The patient will be asked about ADL and practice using the paralyzed upper extremity.
3.2.	The use of the paralyzed upper extremity for ADL will be promoted. The patient who lacked independent practice will be given feedback to improve their motivation.

FMA-UE, Fugl-Meyer assessment of the upper extremity; ARAT, Action Research Arm Test; ADL, activities of daily living.

**Table 2 jcm-13-02212-t002:** Characteristics of analyzed patients.

Characteristics		Female	Male	All
Participants		17 (40)	26 (60)	43 (100)
Age (years)		50 [46, 63]	53 [48, 60]	51 [48, 60]
Height (cm)		158 [154, 164]	170 [166, 173]	166 [161, 171]
Weight (kg)		53 [51, 58]	68 [62, 73]	63 [53, 70]
BMI (kg/m^2^)		21 [20, 22]	23 [23, 25]	23 [21, 25]
Diagnosis	CI	8 (47)	11 (42)	19 (44)
	ICH	9 (53)	15 (58)	24 (56)
Time from onset (months)		139 [105, 172]	122 [100, 165]	133 [100, 167]
Bartel Index		100 [100]	100 [100]	100 [100]
FMA-UE severity	Severe	2 (12)	4 (15)	6 (14)
	Moderate	10 (59)	18 (69)	28 (65)
	Mild	5 (29)	4 (15)	9 (21)

Values are n (%) or median [25th, 75th percentile]. BMI, body mass index; CI, cerebral infarction; ICH, intracranial hemorrhage; FMA-UE, Fugl-Meyer assessment of the upper extremity. FMA-UE total scores of 0–19, 20–46, and 47–66 represented severe, moderate, and mild severity, respectively.

**Table 3 jcm-13-02212-t003:** Raw and delta scores of FMA-UE and ARAT during the study period.

Measurements		Raw	6 m before	3 m before	After IP	3 m after	6 m after
		Delta	-	(1)	(2)	(3)	(4)
FMA-UE	Total	Raw	33.4 ± 12.5	33.2 ± 12.7	31.6 ± 11.1	32.7 ± 11.9	33.3 ± 12.0
		Delta	-	−0.1 ± 2.5	−1.7 ± 4.4	1.1 ± 4.1	0.6 ± 2.5
	Part A	Raw	24.2 ± 6.3	24.3 ± 6.7	23.3 ± 5.9	24.0 ± 5.9	24.3 ± 5.9
		Delta	-	0.1 ± 2.0	−1.0 ± 2.7	0.7 ± 2.0	0.4 ± 1.6
	Part B	Raw	3.5 ± 2.8	3.6 ± 2.8	3.5 ± 2.7	3.4 ± 2.7	3.6 ± 2.6
		Delta	-	0.1 ± 1.1	−0.1 ± 1.3	−0.1 ± 1.2	0.2 ± 0.8
	Part C	Raw	4.3 ± 3.5	4.0 ± 3.3	3.8 ± 3.0	4.0 ± 3.5	4.1 ± 3.6
		Delta	-	−0.3 ± 1.1	−0.3 ± 1.6	0.3 ± 1.9	0.1 ± 1.0
	Part D	Raw	1.3 ± 1.8	1.3 ± 1.7	1.0 ± 1.4	1.3 ± 1.8	1.2 ± 1.8
		Delta	-	0.0 ± 0.6	−0.3 ± 1.4	0.3 ± 1.1	−0.1 ± 0.7
ARAT	Total	Raw	11.1 ± 12.3	10.4 ± 10.9	9.4 ± 11.1	10.0 ± 11.6	10.1 ± 11.8
		Delta	-	−0.7 ± 3.8	−1.0 ± 3.3	0.6 ± 3.7	0.1 ± 1.7
	Grasp	Raw	2.9 ± 4.4	2.8 ± 4.1	2.4 ± 4.3	2.6 ± 4.2	2.6 ± 4.3
		Delta	-	−0.1 ± 1.3	−0.4 ± 2.0	0.2 ± 1.5	0.0 ± 0.7
	Grip	Raw	2.0 ± 2.8	2.0 ± 2.6	1.5 ± 2.7	1.7 ± 2.7	1.7 ± 2.7
		Delta	-	−0.1 ± 0.8	−0.4 ± 1.4	0.2 ± 1.3	0.0 ± 0.6
	Pinch	Raw	1.9 ± 4.3	1.5 ± 3.6	1.4 ± 3.6	1.6 ± 3.8	1.6 ± 3.9
		Delta	-	−0.4 ± 1.8	−0.1 ± 0.9	0.1 ± 1.1	0.0 ± 0.2
	Gross movement	Raw	4.3 ± 1.5	4.1 ± 1.4	4.0 ± 1.4	4.3 ± 1.6	4.2 ± 1.6
		Delta	-	−0.2 ± 0.6	−0.1 ± 0.5	0.3 ± 0.6	−0.1 ± 0.3

Values are mean ± Std. deviation (n = 43). IP, interruption; FMA-UE, Fugl-Meyer assessment of the upper extremity; ARAT, Action Research Arm Test; 6 m before, approximately 6 months before outpatient rehabilitation was interrupted; 3 m before, approximately 3 months before outpatient rehabilitation was interrupted; after IP, after interruption period; 3 m after, approximately 3 months after outpatient rehabilitation was resumed; 6 m after, approximately 6 months after outpatient rehabilitation was resumed; (1) the amount of change in scores from 6 months prior to interruption to 3 months prior to interruption; (2) the amount of change in scores from 3 months before interruption to immediately after resumption; (3) the amount of change in scores from immediately after resumption to 3 months after resumption; (4) the amount of change in scores from 3 months to 6 months after resumption.

## Data Availability

The ethics committee at Jikei University did not authorize the public sharing of the data. Data are available from the Clinical Research Support Center, Jikei University School of Medicine (crb@jikei.ac.jp) for researchers who meet the criteria for access to confidential data. <contact information> Clinical Research Support Center, Jikei University School of Medicine 3-25-8 Nishi-Shimbashi, Minato-ku, Tokyo 105-8461, Japan; Tel.: +81-3-3433-1111 (ext. 2187); Fax.: +81-3-5400-1388.

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
