# Peer review of "Upper-Limb Functional Recovery in Chronic Stroke Patients after COVID-19-Interrupted Rehabilitation: An Observational Study"

_jcm, 2024, doi:10.3390/jcm13082212_

Round 1

Reviewer 1 Report

Comments and Suggestions for Authors

Reviewer comments uploaded. 

Reviewer 2 Report

Comments and Suggestions for Authors

The paper by Daigo Sakamoto et al. is an observational study evaluating the possibility of reversing effects of interrupting rehabilitation treatment following the Covid-19 pandemic. Although the topic may be of interest, the study is burdened by limitations that make it very difficult to support the Authors' conclusions. First of all, the study does not have a control group and this makes it impossible to attribute the improvement in performance to rehabilitation with certainty. In addition, it is not clear how the effects of differences in home rehabilitation self-training can be assessed (i.e. the effects could depend on activities other than those proposed by the occupational therapist), nor is it possible to obtain a dose-response correlation, so that the statement at lines 289-292 is very difficult to support with certainty. In addition to these serious methodological concerns, there are other aspects that require attention: 

- The Authors describe patients with hemiplegia, but there are also patients with mild FMA-UE scores. Perhaps it would be more correct to use the term of hemiparesis?

- No information is given on the comorbidities of the patients, e.g. diabetes, chronic arteriopathy in the lower extremities, nephropathies, which might have limited the activities of the included patients and, consequently, affected their daily practice and limited the effects of the intervention;

- The exercises and activities proposed to the patients included in the study should be specified in much more detail;

- It would be advisable to add, if correct, that the study was conducted in accordance with the Declaration of Helsinki by citing the relevant bibliography on the subject;

- The analyses should be stratified further, differentiating (as they often have different recovery trajectories) ischaemic patients from haemorrhagic patients, taking into account the degree of severity;

- Had spasticity been assessed? How did it affect the results?

- Did the ischaemic patients included receive acute phase treatment for stroke? How long did their hospitalisation last? What about complications during hospitalization? The implications should be considered on the results;

- FMA-UE and ARAT measure different aspects, such as recovery and compensation. The Discussion should be articulated by specifying how much the proposed activities improved ADLs through the use of different strategies or true recovery of motor acts impaired by the stroke. 

Round 2

Reviewer 2 Report

Comments and Suggestions for Authors

I thank the Authors for their extensive work, which greatly improved the quality of their paper.